# An Investigation on Design and Characterization of a Highly Selective LED Optical Sensor for Copper Ions in Aqueous Solutions

**DOI:** 10.3390/s21041099

**Published:** 2021-02-05

**Authors:** Sheng-Chun Hung, Chih-Cheng Lu, Yu-Ting Wu

**Affiliations:** 1Department of Intelligent Automation Engineering, National Taipei University of Technology, Taipei 10608, Taiwan; shengchun@ntut.edu.tw; 2Graduate Institute of Mechatronic Engineering, National Taipei University of Technology, Taipei 10608, Taiwan; tingwu7777@gmail.com

**Keywords:** copper ions, optic sensors, fast response, low cost, high selectivity

## Abstract

The optical characteristics of copper ion detection, such as the photometric absorbance of specific wavelengths, exhibit significant intensity change upon incident light into the aqueous solutions with different concentrations of metal ions due to the electron transition in the orbit. In this study, we developed a low-cost, small-size and fast-response photoelectric sensing prototype as an optic sensor for copper (Cu) ions detection by utilizing the principle of optical absorption. We quantified the change of optical absorbance from infra-red (IR) light emitting diodes (LEDs) upon different concentrations of copper ions and the transmitted optical signals were transferred to the corresponding output voltage through a phototransistor and circuit integrated in the photoelectric sensing system. The optic sensor for copper (Cu) ions demonstrated not only excellent specificity with other metal ions such as cadmium (Cd), nickel (Ni), iron (Fe) and chloride (Cl) ions in the same aqueous solution but also satisfactory linearity and reproducibility. The sensitivity of the preliminary sensing system for copper ions was 29 mV/ppm from 0 to 1000 ppm. In addition, significant ion-selective characteristics and anti-interference capability were also observed in the experiments by the proposed approach.

## 1. Introduction

Recently, there has been lots of water pollution due to industrial drainage, agricultural irrigation and household wastewater, etc. The major water contaminations due to industrial waste and agricultural fertilizer are the hazardous heavy metal ions and other impurities which can directly or indirectly congested into the human body. Soil is also the major environmental compartment which is affected by heavy metals pollution due to the water contaminations. The residence of heavy metals in soils and water is a serious problem because of its presence in food chains. As known, most heavy metals are non-degradable so the damage will last for a long time after being released to the environment. The presence of heavy metals in the environment will decrease the biodegradation rate of organic pollutants, which will double the environmental pollution. On the other hand, lots of metallic products are widely used in daily life, such as food packaging, cookware, cosmetic, herbs, and drinking water supplies [1,2,3]. There are various ways through which heavy metals pose risk to humans, like absorption by plants, food chains, contaminated water and alteration of soil pH value, porosity, etc. The bioaccumulation of heavy metal ion which is over the upper limit in the human body will induce the serious and permanent damage to human organs such as kidney failure, lung dysfunction, osteoporosis, hypertension, stomach ulcer and neuro-disordered [4].

Among the essential trace elements in the human body, such as iron, zinc, manganese and selenium, etc., copper ions (Cu^2+^) take the third place only after iron and zinc because copper ions take part in various fundamental physiological processes. In the human body, copper plays an important part of proteins and enzymes. The content of copper ions in the human body is about 100~150 mg, the concentration of copper in serum is about 100~120 μg/dL. A lot of important enzymes are required for the participation and activation with copper ions, such as tyrosinase, monoamine oxidase, superoxide dismutase and ceruloplasmin, etc. Copper-dependent enzymes are involved in critical biological processes such as synthesis of hemoglobin and elastin, mitochondrial electron transport and metabolism of connective tissue and purine, which ensures that its daily ingestion is indispensable for normal human function [5,6,7]. The copper ions may incorporate into the human body through breathing the air, drinking water, foods, skin contact to the copper material or copper containing compounds. According to the suggestion of World Health Organization (WHO), the upper limit of copper ingestion is 10–12 mg per day and doubled for pregnant women and infants. An excessive copper intake will cause harmful effects, such as irritation of the nose and throat, nausea, vomiting, diarrhea and degenerative heme, and even will induce liver cirrhosis, hepatitis, hypotension, hemolysis, acute renal failure, twitch and death.

There are lots of research groups that have published their results and demonstrated various methods to detect copper ions. Most of them investigated fluorescence sensors for copper ion detection, such as Wang et al. [8], Dong et al. [9], Yu et al. [10], Chen et al. [11], Lee et al. [12], Goswami et al. [13], and Gao et al. [14], etc. The principle of fluorescence sensors employs a light source to excite the sample and to gather the emitted fluorescence by the photodetector. The advantages of fluorescence sensors are high sensitivity and high specificity due to unique optical properties of molecules. However, it also has some disadvantages such as susceptible interference due to pH changes, dependence on oxygen levels and short lifespan of the fluorophore. Alternatively colorimetric analysis using nanoparticles was also used to detect copper ions by lots of group, e.g., Md. Rabiul Awual [15], Yongming Gua et al. [16], and Yu-rong Ma et al. [17]. The colorimetry is a light-sensitive instrument used to determine the absorption and transmittance of light passing through a sample solution. The advantages of colorimetry are low cost, fast response and simple operation by using a spectrometer while it retains some disadvantages like low sensitivity, sample quantity requirements, unrecognizable for colorless compounds, etc. Otherwise, Fan et al. used surface plasmon resonance optical sensor to detect copper ions [18]. Taillades et al. applied ion-selective-electrodes (ISE) and ion-sensitive field-effect transistor (ISFET) microsensors to detect copper ions in solution [19]. The principle of the ion-selective electrode is the transport of ions from a high concentration to a low one through a selective binding with some sites within the membrane that creates significant potential difference. The ISE sensors have the advantages like linear response, non-destructive sensing of analytes, short response time and independence on color or turbidity. Nevertheless, there are still some limitation for ISE sensors, such as limited shelf life of the electrodes, low sensitivity, contamination issues of the electrode by proteins or other organic solutes, etc. All the investigations mentioned above show their latest results in detecting copper ions; however, the sampling procedure is complex and costly instruments are required. For the above reasons, it is difficult to apply metal ion detection solutions in daily life such as in-situ monitoring for house-held drinking or agricultural irrigation water.

In the present work, we demonstrate a low-cost, small-size, highly selective photoelectric sensing prototype which could be used for in-situ monitoring the copper ions in aqueous solution. The photoelectric sensing system was integrated with optoelectronic elements for measuring the absorbance of specific wavelength absorption, LED light sources with a specific wavelength band, two cuvettes which contain reagent and reference solutions and designated amplifier circuit. The photoelectric sensing system will measure and quantify the absorbance of LED incident light in terms of the voltage output of phototransistors to translate the concentration variation of copper ions in aqueous solution. More importantly, the photoelectric sensing system will survey specificity and interference for copper ions in the aqueous solutions while cadmium ions, nickel ions, iron ions, and chloride ions exist simultaneously. A preliminary experiment on reproducibility and stability with respect to dynamic ion concentrations is also investigated.

## 2. Materials and Methods

### 2.1. Preparation of Solutions

High-purity, deionized water purified by Milli-Q water purification system (Millipore Elix 3 and Milli-Q Academic Gradient A-10) was used for preparation of reagents and standards. The samples and standards were quantified by analytical pipette from Gilson Incorporated (P.O. Box 620027, Middleton, WI, USA) and the final volumes were measured by mass in all cases. The standard solutions for calibration were prepared from a stock solution of 1000 mg/L by successive dilutions with Milli-Q deionized water. Calibration standards (Cu, Cd, Ni, and Fe and Cl) were purchased from High-Purity Standards (P.O. Box 41727, Charleston, SC, USA) and Thermo Scientific (Waltham, MA, USA), respectively. The contents and the catalogue no. of the standard solutions were shown in Table 1.

### 2.2. Optical Sensing Principle

In this study, an UV/Vis spectrophotometer (LKU-5100, TiHalinko Technology CO., Taipei, Taiwan) with quartz cuvette were utilized to measure the optical characteristics such as absorbance of the aqueous solution measured by different concentrations of metal ions. The spectrophotometer was interfaced with computer, which was operated by UV-professional analysis software. According to the Beer–Lambert law, which relates the radiant power in a beam of electromagnetic radiation, such as ordinary light, to the length of the path of the beam in an absorbing medium and to the concentration of the absorbing species, respectively. The laws are normally described in the Equation (1) as follows:(1)$$A=-\log_{10}\frac{P}{P_{0}}=\mathrm{abc}$$
where A = absorbance, P = transmitted light intensity, P_0_ = incident light intensity, a = absorptivity, b = length of the beam in the absorbing medium, and c = concentration of the absorbing species. This law, vital to colorimetric analysis and spectrophotometry, can be extended to the absorption of different wavelength because of the energy of the electron transition in the orbit. In this work, we quantify the concentration of copper ions in the aqueous solution by utilizing the relation of optical absorbance with specific incident wavelength according to the electron transition. Due to the difference of the electron orbits with different metal ions, the spectrum of the absorbance will show different peaks or bands. The aqueous solutions with other metal ions, such as cadmium, nickel, iron and chloride ions, were measured. We also particularly investigated the measured results about the specificity of copper ion detection using light absorption according to the Beer–Lambert law in this work.

### 2.3. Design and Setup of the Photoelectric Sensing System for Copper Ion Detection

The proposed sensing technique, compared with other methods, provides a relatively simple structure to detect different concentrations of copper ions in aqueous solution by utilizing the electron transition principle. Usually the instrumentation equipment for detecting optical signals is normally massive and not portable. However, with the function of the UV/Vis spectrophotometer, a portable photoelectric sensing system was designed and fabricated. As shown in Figure 1, the light sources (LEDs with specific wavelength at 850 nm for this study), the detection elements (Si based NPN phototransistor with model number BPW77N, manufactured by Vishay Telefunken, Heilbronn, Germany), the cuvettes which can fill in the reagents with different concentrations and the reference (deionized water), the printed circuit board (PCB) board with a differential amplifier circuit and a voltage follower on it, the power supply and the multi-meter were all integrated with the optical sensor. In order to prevent the external light source from affecting the sensitivity and accurate analysis of the copper ion detection in the optical sensor system, we use opaque acrylic as a carrier to fix the LED, liquid sample and phototransistor. Therefore, the influence of external light sources on the experiment can be minimized.

### 2.4. Design of the Circuit

In this paper, we demonstrate a simple way to detect the metal ions in the aqueous solution using light absorbance by a Si based NPN phototransistor. The output electrical signal was processed by the circuit as shown in Figure 2. V_D_ is the output voltage of the phototransistor which is next to the regents with different concentration of metal ions and V_R_ is the output voltage of the phototransistor which is next to the DI-water reference. Since the input and output voltage are identical in the voltage follower, so:*V_D_* = *V*_1_, *V_R_* = *V*_2_(2)

And we set
*R*_1*a*_ = *R*_1*b*_ = *R*_1_, *R*_2*a*_ = *R*_2*b*_ = *R*_2_, *V_a_* = *V_b_*(3)

So,
(4)$$V_{\mathrm{out}}=\;V_{a}+\;V_{R2a},\;i_{D}\;=\;\frac{V_{1}}{R_{1a}},\;\;i_{R}\;=\frac{V_{2}}{\;R_{2b}}$$

By voltage divider rule,
(5)$$V_{a}\;=\;i_{R}\cdot R_{2b}$$

According to the inverter,
(6)$$V_{R2a}\;=\;-\frac{R_{2a}}{R_{1a}}\cdot V_{D}$$

So,
(7)$$V_{\mathrm{out}}\;=\;V_{a}+V_{R2a}\;=\;i_{R}\cdot R_{2b}-\frac{R_{2a}}{R_{1a}}\cdot V_{1}\;=\;\frac{R_{2b}}{R_{1b}}\cdot V_{2}-\frac{R_{2a}}{R_{1a}}\cdot V_{1}\;=\;\frac{R_{2}}{R_{1}}\cdot \left(V_{2}-V_{1}\right)$$

## 3. Results

### 3.1. Photometric Interpretion of Light Absorbance of Copper Ions in Aqueous Solution

The aqueous solutions used in this study were prepared by 1000 ppm Cu in 2% HNO_3_ standard solution from High-Purity Standards and diluted with Milli-Q deionized water (Merck KGaA, Darmstadt, Germany) into different concentrations. LKU-5100 UV/Vis spectrophotometer (TiHalinko Technology CO., Taipei, Taiwan) was utilized to measure the absorbance of aqueous solutions with incident light wavelength from 200 to 1000 nm. As shown in Figure 3a the spectrum of absorbance shows two bands at 307 nm and 809 nm, respectively. According to the previously published work, the UV-Vis absorption bands of monovalent copper are due to the 3d^10^→3d^9^4s and 3d^10^→3d^9^4p transitions [20,21,22], so the Cu^+^-Cu^+^ dimers will contribute to the absorption in the range of 280–330 nm region. On the other hand, the free divalent copper ion has the 3d^9^ electronic configuration and the ground state being ^2^D. In the octahedral field of the matrix, it is the ^2^E_g_ level that corresponds to the ground state of the ion. Due to the tetragonal distortion, we can expect three d-d transitions in the Cu^2+^ ion instead of a single one, which results in a broad absorption band of divalent copper with a maximum around 750–950 nm [23,24]. In this work, as shown in Figure 3a, there are two bands of the optical absorption observed around 307 nm and 809 nm which are from the optical absorption of Cu^+^-Cu^+^ dimers and Cu^2+^ ions, respectively. Moreover, the standard solution used in this study is copper ions in 2% HNO_3_, and according to the research by Antonio Febo et al. [25] and A. L. Goodman et al. [26], there will be a gas-phase HONO in the aqueous solution of HNO_3_ and will absorb strongly in the UV band at 310 nm. As shown in Figure 3b, the aqueous solutions with and without copper ions both show the absorption peak around 310 nm and the one with copper will be stronger, which indicates both the gas-phase HONO and Cu^+^-Cu^+^ dimers absorb the incident light around 310 nm and the results will be interfered each other. So it is difficult to quantify the concentration of copper ions using UV/Vis spectrophotometer around 310 nm vicinity. On the other hand, the broad band of the absorbance change due to the Cu^2+^ ions exhibits excellent specificity with the gas-phase HONO and the absorbance of the aqueous solution increased with the rising concentrations of copper ions at incident light wavelength around 809 nm, as shown in Figure 3c. Therefore, we designate to quantify the optical absorbance due to the electron transition of Cu^2+^ ions in the aqueous solution, which will contribute the broad absorption band around 809 nm in UV/Vis spectrophotometer. The results of the photometric measurement reflect very excellent linearity with increasing copper concentrations, as shown in Figure 3d, which can be interpreted via the Beer–Lambert law due to the three d-d electron transition of Cu^2+^ ions.

### 3.2. Experimental Results of Photoelectric Sensing System

In fact, based on the Beer–Lambert law, the intensity of light will decay proportionally with increasing analyte concentrations, and there exists inversely linear relation between them. In order to transfer the optical intensity signal into electrical signal, a phototransistor (BPW77N, Heilbronn, Germany) with narrow viewing angle was utilized. According to the results of UV/Vis spectrophotometer, the absorption band around 809 nm vicinity shows excellent specificity toward copper ions. As shown in Figure 1, to in-situ correct the decay of the light intensity from the LEDs, two identical LEDs were employed to calibrate the signal of the absorption. The two identical and commercially available LEDs as the light source which emit light with 850 nm wavelength into the reagent solution with different copper ion concentrations and deionized water at the same time. The output voltage of two phototransistor were introduced into the operational amplifier. According to Equation (7), the *V*_out_ will be proportional to the difference of the voltages from the phototransistors measuring the transmitted light through the reagent solution and reference solutions. The higher concentration of metal ions in the solution is present, the lower transmitted light intensity is detected, which will contribute a higher voltage. The output electrical signals processed by the circuit in Figure 2 were test and quantified. As shown in Figure 4, the output voltage of the circuit still shows good linearity with increasing copper ion concentration and corresponds to the results of optical absorbance. Comparing Figure 3D and Figure 4, it can be inferred that the electrical output response show less linearity with the ion concentration variation than the photometric output signal and it is because of the intrinsic noise from the transducing circuit and the potential decay of incident light wavelength from the LEDs light source. However, the experimental results of the as-designed cost-effective system still show that we can quantify the concentration of copper ions instead of using the spectrophotometer and obtain satisfactory linearity with copper ion concentration. According to Figure 4, the sensitivity of the copper ion sensor were estimated as 29 mV/ppm.

In addition, the photoelectric sensing system of optic sensor for copper ions also showed a good reproducibility, as illustrated in Figure 5. The concentration of the aqueous solution with copper ions were switched from 0 ppm (DI-water) to specific upper limits such as 100, 200, 300, 400, 500 and 1000 ppm, respectively, for several times. As can be seen in Figure 5, the change of the electrical output voltage between 0 ppm and 100 ppm was considerably larger than the background noise and the optic sensor for copper ions exhibited a satisfactory operation in terms of stability and reproducibility.

Since the aqueous solution may be composed of lots of impurity and metal ions, it is extremely important to have high specificity toward specific metal ions of the sensors. In this investigation, the specificity of the optic sensor for copper ions were also carried out. In order to measure the specificity of the sensors for copper ions, the reagents were made up of different metal ions from the standard solutions, such as Cd, Ni, Fe, and Cl ions. The contents of the standards were shown in Table 1. As shown in Figure 6a, the results of UV/Vis spectrophotometer all showed that there were strong absorption peaks occurred around 310 nm vicinity in the aqueous solutions with 1000 ppm of Cd, Ni, Fe, Cl and Cu ions because the standard solutions were made with 2% HNO_3_, which will form the gas-phase photolysis of nitrous acid (HONO). However, the broad band absorption around 800 nm only occurred in three d-d electron transition of Cu^2+^ ions. Although the absorption band is very broad, however, the absorbance of all the other aqueous solutions around 800 nm vicinity indicates an order of magnitude smaller than that of copper ion solution, which reveals excellent specificity for copper ions. The same results were also observed in the electrical output response by the photoelectric sensing system, as shown in Figure 6b. The output voltage of the sensor with the incident light wavelength at 850 nm specified a very low response to each of these specific metal ions in solutions when compared to Cu ions, which means the incident light wavelength at 850 nm can be an effective absorption band for Cu ions detection. Good sensitivity and linearity with only Cu ions solution were obviously observed and well-maintained.

In order to study the interference phenomenon caused by the diverse metal ions in the same solution, the sample solution was made with various concentrations of copper ions from 100 ppm to 1000 ppm and each solution was mixed with 200 ppm aqueous solutions of Cd, Ni, Fe, and Cl ions, respectively. Through the spectrum of the spectrophotometer, we observed that the absorbency of the mixture for the specific ions is very close to that of the copper ion sample, which is not shown here. Hence, we also verified the electrical output voltage of the circuit of the sensing system with respect to the specific ion mixture. Experimental results are shown in Figure 7a–d, and it can be seen that the output voltage resulting from the sensor system demonstrates a similar trend of sensitivity and linearity change with the copper ion sample. On the other hand, the results also illustrate that chemical interaction of Cd, Ni, Fe, and Cl ions with Cu ions are not significant due to the equivalent or slightly changed slope of the calibration curves in these figures, though these additional ions apparently are likely to interfere the three d-d electron transition around 850 nm wavelength. To explain the difference between the photometric and photoelectric measurements, it is because the incident light from UV/Vis spectrophotometer is made of a single beam with grating resolution of 1200 lines/mm and wavelength accuracy of ±1 nm while the light source of the low-cost photoelectric sensor is IR LEDs with broad emission band from 800 to 875 nm and the peak position is 850 nm. There should be some extra absorbed light intensity noise at the wavelength instead of 850 nm, which is probably the reason why the interference phenomenon in the photoelectric experiment was observed. In spite of the broad information of light absorption band around 850 nm vicinity, this approach proposed in this study still provides excellent specificity and strong anti-interference capability for detecting copper ions in aqueous solution. We also compare our results with those of previous research teams, as shown in Table 2. This proves that our optical sensor has the characteristics of fast detection speed and low price.

## 4. Conclusions

In summary, an in-situ photoelectric sensing system as an optic sensor for copper ions detection is successfully demonstrated. The cost-effective, small-size and fast-response prototype is carried out by mainly evaluating the change of photometric absorbance from IR LEDs with different concentrations of copper ions, transferring the optical signals to the corresponding electrical voltage through a phototransistor, and implementing a transducing circuit incorporated in the sensor. It is shown that the highly selective optical sensor for copper ions can be viable by employing the broad absorption band around 850 nm which is contributed by the three d-d electron transition of Cu^2+^ ions. The optical absorbance signal can be successfully quantified and transferred into electrical output voltage by the photoelectric prototype. The sensitivity of as-designed optic sensor was found 29 mV/ppm and a linear working range up to 1000 ppm of Cu ions is performed. The optic sensor for copper ions also demonstrated both good reproducibility and excellent resistance to interference impurities. These results from the investigation may offer a potential for extensive and real-time applications in portable or house-held metal ion detection needs.

## Figures and Tables

**Figure 1 sensors-21-01099-f001:**
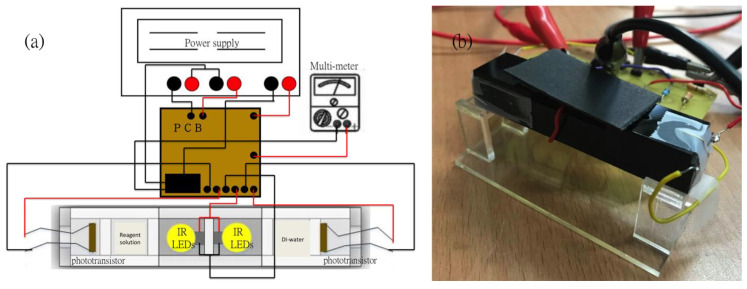
(**a**) The schematic of the photoelectric sensing system; (**b**) The real image of the photoelectric sensing system.

**Figure 2 sensors-21-01099-f002:**
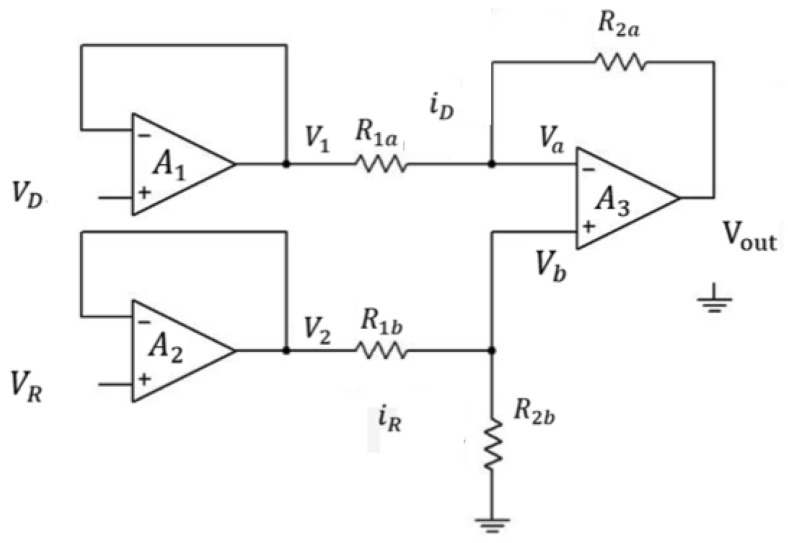
Layout schematic of the amplifier circuit of the optical sensor.

**Figure 3 sensors-21-01099-f003:**
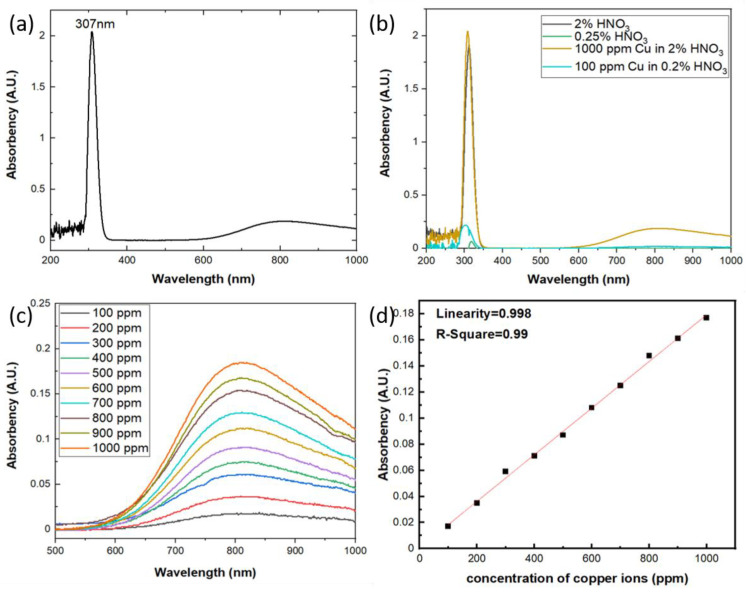
(**a**) The absorption spectrum of aqueous solution with concentration 1000 ppm of copper ions; (**b**) the absorption spectrum of aqueous solution with HNO_3_ and copper ions; (**c**) the absorption spectrum of aqueous solution with different copper ion concentrations at 809 nm; (**d**) The relation of the optical absorbency versus the concentration of copper ions.

**Figure 4 sensors-21-01099-f004:**
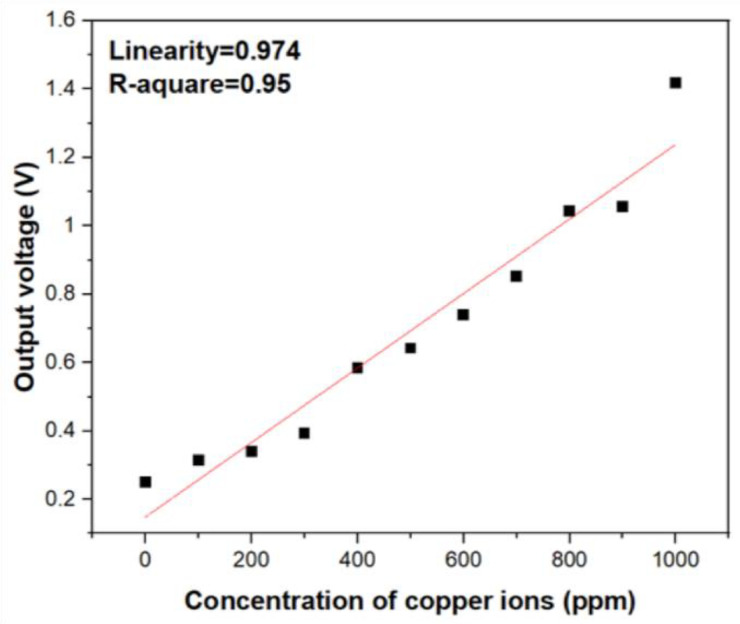
The electrical output voltage versus concentration variation of copper ions.

**Figure 5 sensors-21-01099-f005:**
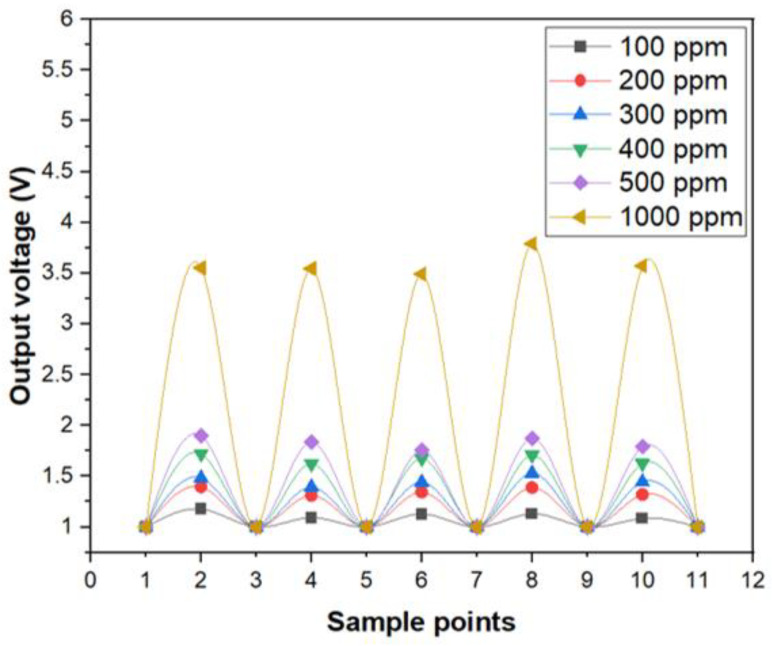
The reproducibility test of optic sensor for Cu ions.

**Figure 6 sensors-21-01099-f006:**
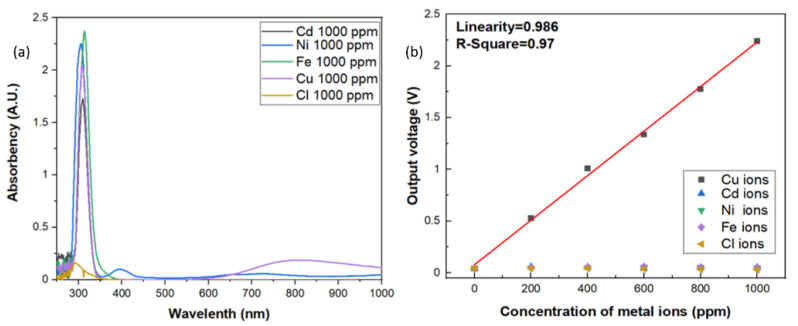
(**a**) Specificity test of the optic sensor for a series of ions. (**b**) The electrical output response to different ion solutions.

**Figure 7 sensors-21-01099-f007:**
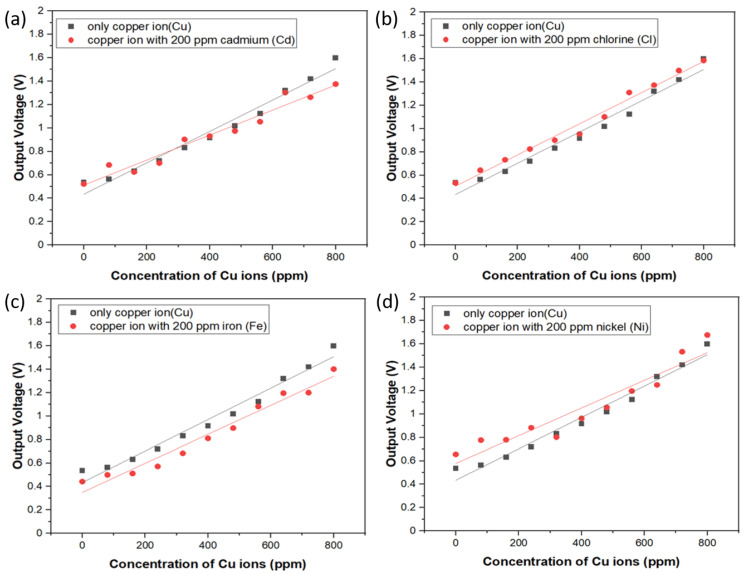
Interference tests of electrical output voltage with respect to diverse metal ion sets. (**a**) copper ions only and copper ions with 200 ppm Cd; (**b**) copper ions only and copper ions with 200 ppm Cl; (**c**) copper ions only and copper ions with 200 ppm Fe; (**d**) copper ions only and copper ions with 200 ppm Ni.

**Table 1 sensors-21-01099-t001:** The catalogue no. of the calibration standards.

Element	Calibration Standards	Catalogue No.
Cu:	1000$\;\pm \;3\;\mu g/\mathrm{mL}\;\mathrm{in}\;2{\%\;\mathrm{HNO}}_{3}$	100014-1
Cd:	1000$\;\pm \;3\;\mu g/\mathrm{mL}\;\mathrm{in}\;2{\%\;\mathrm{HNO}}_{3}$	100008-1
Ni:	1000$\;\pm \;3\;\mu g/\mathrm{mL}\;\mathrm{in}\;2{\%\;\mathrm{HNO}}_{3}$	100036-1
Fe:	1000$\;\pm \;3\;\mu g/\mathrm{mL}\;\mathrm{in}\;2{\%\;\mathrm{HNO}}_{3}$	100026-1
Cl:	Sodium chloride in water	941708

**Table 2 sensors-21-01099-t002:** Comparison table of recent research results.

Sr. No.	Measurement Techniques	Analytical Ranges (Detection Limit) (N/A: Not Announced)	Selectivity	Response Time	Cost	Ref.
1.	Fluorescent Sensors	2.5–35 μM (1.8 μM)	Good	8 min	N/A	[8]
2.	Fluorescent Sensors	N/A (12.7 ppb)	Good	N/A	N/A	[9]
3.	Fluorescent Sensors	N/A (2.32 × 10^−5^ M)	Good	N/A	N/A	[10]
4.	colorimetric chemosensor	280–800 nM (0.15 μM)	Good	N/A	N/A	[12]
5.	fluorescence	N/A (2.3 μM)	Good	N/A	N/A	[13]
6.	Colorimetric detection	N/A (0.10 μg/L)	Good	30 min	N/A	[15]
7.	Colorimetric detection	N/A (20 μM)	Good	N/A	N/A	[16]
8.	Colorimetric detection	0.03 μM	Good	9 min	N/A	[17]
9.	fluorescent chemosensor	1.0 × 10^−7^~2.5 × 10^−6^ mol/L (2.0 × 10^−8^ mol/L)	Good	N/A	N/A	[18]
10.	ISE and ISFET microsensors	1.0 × 10^−6^ mol/L	Good	10 s	N/A	[19]
11.	Optical sensor	1~1000 ppm	29 mV/ppm	Less than 5 sec	Low (less than 40 USD)	This work

## Data Availability

No new data were created or analyzed in this study. Data sharing is not applicable to this article.

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
