# Peer review of "An Investigation on Design and Characterization of a Highly Selective LED Optical Sensor for Copper Ions in Aqueous Solutions"

_sensors, 2021, doi:10.3390/s21041099_

Round 1

Reviewer 1 Report

The manuscript “An investigation on design and characterization of a highly selective LED optical biosensor for copper ions in aqueous solutions” by Hung et al.,  presents a LED optical sensor for detection of Cu-ions in water.

I have several questions on this work

  1. Why it is a biosensor? No biological material is used for detection purposes. More appropriate will be to modify biosensor to sensor
  2. Why Cu concentration is expressed in mV? In Abstract Limit of detention is expressed as mV/ppm
  3. How water sampling was done? Which volume was used in detections? What volum is recommended by the official method?
  4. This sensor should be compared with previously developed. For instance, the concentration of Cu-ions can be easily and with a high sensitivity detect by an electrochemical method. The advantages of the new method should be explained by adding a comparative table in Discussion showing time of analysis, limit of detection, price…
  5. Can this method be applied in waste waters that are not fully transparent?

Reviewer 2 Report

The article has highlighted the in-situ mechatronic sensing system and optical biosensor and furthers its application in copper ion detection. The authors have demonstrated the successful quantification of electrical output voltage also the biosensor has shown good reproducibility and resistance to an inference of impurities.

My only comment is to remove the word " Opto-mechatronic system" as such there is no mechanical mechanism is involved. 

Reviewer 3 Report

The presented research demonstrated opto-mechatronic sensing system as a point-of-care testing (POCT) device for evaluating the copper ions by assembling the portable electric components. The developed optical sensor enables quantitative analysis of Cu in aqueous solution containing various metal ions. After careful reading of the original research, I recommend the manuscript could be revised before publication, and the details are below.

  1. To sensitively and accurately analyze the Cu based on optical sensing system, the test should be implemented in the dark condition. How did you prevent the ambient light? The detail experimental condition should be included.
  2. The real images of developed opto-mechatronic sensor should be added.
  3. In the Fig. 4, the clinical significance such as safe range or harmful range for Cu calibration curve should be explained.

Round 2

Reviewer 1 Report

The authors have responded to my comments. 

Author Response

Thank you!